# Immunohistochemical expression of substance P in breast cancer and its association with prognostic parameters and Ki-67 index

**Maha S. Al-Keilani**[1]* , **Rana I. Elstaty**[2] , **Mohammad A. Alqudah**[3] , **Asem M. Alkhateeb**[2]

**1** Department of Clinical Pharmacy, College of Pharmacy, Jordan University of Science and Technology, Irbid, Jordan, **2** Department of Biotechnology and Genetic Engineering, College of Science and Art, Jordan University of Science and Technology, Irbid, Jordan, **3** Department of Pathology and Microbiology, College of Medicine, Jordan University of Science and Technology, Irbid, Jordan

☯ These authors contributed equally to this work.

* mskeilani@just.edu.jo

**Data Availability Statement:** All relevant data are within the paper and its Supporting information files.

## Abstract

### Background

The neuropeptide substance P is a potential biomarker and therapeutic target in cancer. The main objectives of this study were to investigate the expression level of substance P in different breast cancer molecular subtypes and identify its association with clinicopathological parameters of patients and with Ki-67 index.

### Methods

A retrospective analysis was performed for a total of 164 paraffin-embedded breast cancer tissue samples [42 Her2/neu-enriched, 40 luminal A, 42 luminal B (triple-positive) and 40 triple negative subtypes]. The tissue microarray slides containing specimens were used to determine the expression of substance p and Ki-67 by immunohistochemical staining.

### Results

The mean age of the cohort was 51.35 years. Twenty two percent of cases had low substance P expression levels (TS $\leq$ 5), while 78% had high expression levels (TS > 5). A significant association was found between SP expression level and breast cancer molecular subtype (p = 0.002), TNM stage (p = 0.034), pN stage (p = 0.013), axillary lymph node metastasis (p = 0.004), ER and PR statuses (p<0.001) and history of DCIS (p = 0.009). The average percentage of Ki-67 expression was 27.05%. When analyzed as a continuous variable, significant differences were observed between the mean Ki-67 scores and molecular subtype (p = 0.001), grade (p = 0.003), pN stage (p = 0.007), axillary lymph node metastasis (p = 0.001), and ER and PR statuses (p <0.001).

**Funding:** MSA received a fund by the deanship of research at Jordan University of Science and Technology (grant number: 2018266). The funders had no role in study design, data collection and analysis, decision to publish, or preparation of the manuscript.

**Competing interests:** The authors have declared that no competing interests exist.

**Abbreviations:** DCIS, ductal carcinoma in situ; EGFR, epidermal growth factor receptor; ER, estrogen receptor; Her2/neu, human epidermal growth factor receptor 2; IHC, immunohistochemistry; IS, intensity score; NK1R, neurokinin 1 receptor; PR, progesterone receptor; PS, proportion score; SP, substance P; TMA, tissue microarray; TNBC, triple negative breast cancer; TS, total score; WHO, world health organization.

## Conclusion

SP is overexpressed in most of the analyzed tissues and has a negative prognostic value in the breast cancer patients. Besides substance P is a potential therapeutic target in breast cancer.

## Background

Breast cancer is the most common cancer among women worldwide accounting for 30% of all cancer cases [1]. Heterogeneity is a hallmark for breast cancer at several levels including histological and morphological features, immunohistochemical profiles, clinical presentation, and response to therapy [2]. Histologically there are at least 17 different types of breast cancer where invasive ductal carcinoma accounts for 50%-70% of invasive cases [2, 3]. Breast cancer has been also subtyped based on the molecular profile into four major groups based mainly on the expression status of hormone receptors; estrogen receptor (ER) and progesterone receptor (PR), and Her2/neu status. These molecular subtypes are luminal A (ER+ve, PR+ve, Her2/neu-ve), luminal B (triple positive), Her2/neu-enriched (ER-ve, PR-ve, Her2/neu+ve) and basal like (triple negative breast cancer; TNBC) [4].

The treatment of breast cancer includes surgery, radiotherapy, chemotherapy, hormonal therapy and targeted therapy directed to Her2/neu [3]. However, about 15% of breast cancer cases are TNBC [4], where hormonal and targeted therapies are not effective [5], thus representing an aggressive type of breast cancer that requires the identification of new therapeutic targets.

Ki-67, a nuclear protein, is used as a proliferation marker in breast cancer tissues [6–8]. It has a molecular weight of about 359 KDa and it presents at all cell cycle phases except the G0 phase [9]. The function of Ki-67 is still unknown, but it was shown to play an important role as a prognostic and predictive marker in breast cancer [9–11].

Substance P, an undecapeptide protein of the tachykinin family of sensory nerve neuropeptides, is widely expressed in the nervous and immune systems and is being studied as a potential prognostic biomarker and therapeutic target in cancer. SP was overexpressed in different types of cancers such as colorectal, pancreatic, breast and oral squamous cell carcinoma [12–19]. Upon preferential binding to the neurokinin 1 receptor, SP participates in several vital carcinogenesis processes including cancer cell proliferation, survival, angiogenesis and metastasis [13, 16, 20–24]. Furthermore, blockage of NK1R via utilizing receptor antagonists resulted in the reversal of the SP-induced tumorigenic effects in vitro and in vivo [20, 22, 23, 25, 26]. SP also possessed oncogenic roles in breast cancer; it was overexpressed in breast cancer cell lines, facilitated bone marrow metastasis, and resulted in the transactivation of various receptors with tyrosine kinase activity such as Her2/neu and EGFR, thus enhanced breast cancer malignancy and metastasis and resistance to anticancer therapy [15, 19, 27, 28].

Up to our knowledge, this is the first study to report the differential expression of SP in the four major breast cancer molecular subtypes. The aims of this study were to evaluate the expression level of SP and its clinical significance in breast cancer patients via investigating the potential association with clinicopathological parameters. Moreover, the relationship between SP and the proliferative marker; Ki-67, was assessed.

## Materials and methods

### Patients

This is a retrospective analysis for a total of 164 paraffin-embedded breast cancer tissue samples provided by the department of pathology of King Abdulla University Hospital (KAUH) in

Irbid, Jordan. The samples represented female patients with stages I to IV who underwent surgical resection between 2007 and 2019; and did not receive chemotherapy or radiotherapy prior to surgery. Clinicopathological data including age, gender, tumor volume, degree of histological differentiation (well/moderate/poor, WHO), depth of infiltration, staging, and status of lymph nodes and distant metastasis, and hormone receptor (ER and PR) and Her2/neu statuses were retrospectively collected from patients' medical charts. A 5% staining proportion was used as the cut-off point for both ER and PR status by IHC. Her2/neu was evaluated according to the guidelines of IHC, which consider 0 or +1 as negative and +3 as positive. Fluorescence In Situ Hybridization (FISH) was performed for tissues with IHC +2 value and based on the results they were classified as positive or negative. Out of the 164 patients, 42 were Her2/neu-enriched, 40 were luminal A, 42 were luminal B (triple-positive) and 40 were triple negative (TNBC).

## Ethics approval and consent to participate

This study was approved by the Institutional Review Board (IRB) at Jordan University of Science and Technology with IRB approval reference number 28/116/2018. All data obtained were fully anonymized and because of the retrospective nature of study the IRB waived the requirement for informed consent.

## Tissue microarray (TMA) construction and TMA slide preparation

The breast cancer formalin fixed paraffin-embedded blocks were chosen based on the molecular subtype for each case, and then cancerous areas of breast tissues were selected and marked on the identical hematoxylin and eosin (H/E) slide and sampled for TMA blocks. With a tissue-microarrayer (3DHISTECH TMA master II), TMAs were prepared using a 2.0mm thin puncher needle. TMA Master II creates holes in the recipient paraffin block to insert "home" tissue cores from the donor block. One core of tumor cells for each sample was transferred from the donor block to the recipient block. An X-Y position guide for the recipient block was automatically adjusted by the TMA control software application. Eight tissue array blocks were constructed to include the entire 164 cores of interest in addition to control cores. Colon tissues that are known to express Ki-67 protein were used as a positive control for Ki-67, lung tissues were used as a positive control for SP, and normal breast tissues were used as a negative control. The tissue microarray slides containing specimens were used to determine the expression of Ki-67 and SP by immunohistochemical staining (IHC). TMA sections of (2 μm) thickness were mounted on coated slides using Accu-Cut® SRM™ 200 Rotary Microtome and prepared for IHC.

## Immunohistochemical staining (IHC)

The automated Ventana Bench Mark ULTRA IHC/ISH Staining Module (Ventana Co., Tucson, AZ, USA) was used together with ultraView universal DAB (3' diaminobenzidine) IHC detection method (Ventana Co., Tucson, AZ, USA) on the 2 μm tissue sections of TMA slides. The primary antibodies used were as follows: anti-SP (1:50, Abcam, Cat# ab10353, RRID: AB_297089) and anti-Ki-67 (prediluted, Ventana Medical Systems, Cat# 790–4286, RRID: AB_2631262).

## Staining evaluation

All findings were evaluated by two independent pathologists who had no knowledge of the patients' clinicopathological data to avoid bias. A light microscope (Olympus Corporation,

Japan) was used to visualize the slides. Ten visual fields were selected for each slide and examined at 40× magnification, and pictures were obtained by a PC-driven digital camera (Olympus DP74, Japan).

## Immunohistochemical analysis and scoring

A semi-quantitative scoring system (Allred unit scoring system) was used for IHC evaluation of SP, based on a combination of the proportions of positively stained cells (PS) and the intensity of the staining (IS). The IS was as follows; 0, no staining; 1, weak staining (light yellow); 2, moderate staining (yellow brown); and 3, strong staining (brown). PS ranged from 1%-100%, given as follows: 0, 0% reacting cells; 1, <1% reacting cells; 2, 1%-10% reacting cells; 3, 11%-33% reacting cells; 4, 34%-66% reacting cells; and 5, >= 67% reacting. After that the two scores were added together for a total score (TS) with eight values. Tissues were divided into two groups based on the TS. Tissues with a TS of ≤5 were considered as low expression of SP, while those with a TS of >5 were considered as high expression of SP. Nuclear staining of SP was evaluated. The Ki-67 index was obtained by the percentage of total number of tumor cells with nuclear staining. Representative images from well preserved areas in breast cancer tissues are demonstrated in Fig 1 for SP.

## Statistical analysis

Data were collected in an Excel database from Windows 10 (Microsoft Corporation, Redmond, WA, USA) and SPSS statistical software system (IBM SPSS Statistics 23, USA) was used

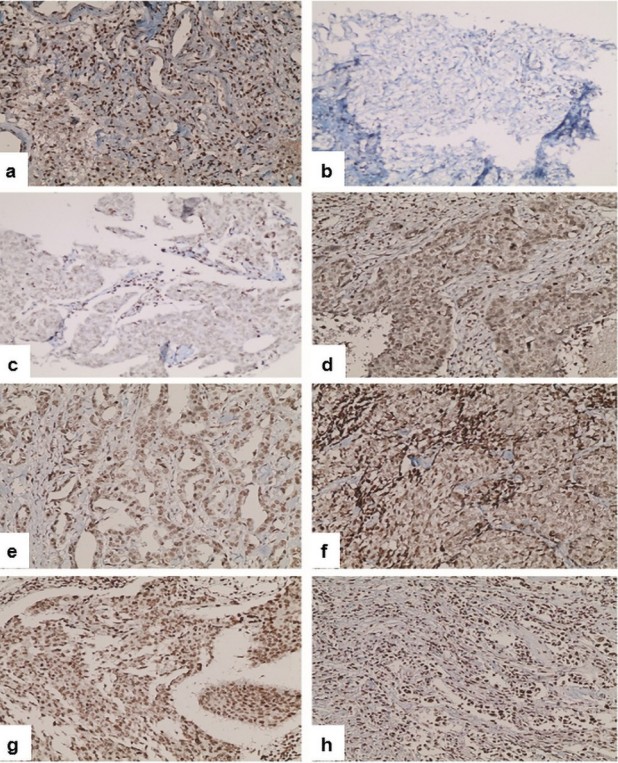

**Fig 1. Immunohistochemical staining of SP in breast cancer tissues and positive control.** (a) Positive control (lung tissues). (b-h) Different SP expression total scores (TS) in breast cancer tissues from 0–8. (b) Score 0. (c) Score 3. (d) Score 4. (e) Score 5. (f) Score 6. (g) Score 7. (h) Score 8. Original magnification 20x.

for statistical analyses. Descriptive statistics were done. Pearson Chi-square ($\chi^2$) test was used to compare the staining scores between groups. ANOVA and student's t-test were used to compare group means. Tukey post hoc test and Bonferroni Chi-Square residual analyses were used for multiple comparisons as appropriate. Pearson correlation test was used to investigate the correlation between different parameters of the studied groups. Continuous variables were presented as mean ± standard deviation, while categorical variables were presented as numbers and percentages. Statistical significance was considered if p <0.05. Bonferroni-adjusted p-value of $0.05/n_{tests}$ were used as threshold for significance.

The complete dataset of clinical, pathological, and staining data can be found in S1 File.

## Results

### Clinical and pathological characteristics

This study was carried out on 164 retrospective breast cancer cases received in the pathology department of King Abdulla University Hospital (KAUH) in Irbid, Jordan from 2007 to 2019. All samples were from female patients.

The cases were from four breast cancer molecular subtypes; 42 were Her2/neu-enriched, 40 were luminal A, 42 were luminal B (triple-positive) and 40 were triple negative (TNBC). The average age (±SD) of the cohort was 51.35 (±11.2) years with a range of 28–82 years at the time of surgery. Whole pathological reports were obtained for all selected patients. Table 1 shows the major demographic and clinicopathological characteristics of our cohort. It is shown that all cases were invasive ductal carcinoma. Tumor volume was calculated using the ellipsoid model formula: *Tumor volume = π/6 (a × b × c)*, where a, b and c represent three perpendicular diameters. The mean tumor volume in cubic centimeter (±SD) of the cohort was 38.97 (±67.41) $cm^3$ with a range of 0.18–571.77 $cm^3$. About 67% of the cases were grade III, 28% were grade II and the remaining were grade I. Of total cases, 4.3% were stage I, 27.4% were stage II, 34.1% were stage III and 32.3% were stage IV. More than half of the cases were pT2 accounting for 53.7%. Regarding lymph nodes metastasis (pN stage), 26.8% were pN0, 23.8% were pN1, 20.1% were pN2 and 26.2% were pN3. 107 cases had no evident distant metastasis (M0) and 54 cases had distant metastasis (M1). Axillary lymph node metastasis was positive in approximately 71% of cases and lymphovascular invasion was evident in 57.3% of cases. More than two third of cases (68.9%) had a positive personal history of ductal carcinoma in situ (DCIS). 29 cases had a known positive family history of breast carcinoma.

### Immunohistochemical findings

As shown in Table 1, 36 cases (22%) had low SP expression levels (TS ≤ 5), while 128 cases (78%) had high expression levels (TS > 5).

The average percentage of Ki-67 expression was 27.05 ± 26.86. The expression ranged from 0% to 95%. Fig 2a shows the expression level of Ki-67 with frequencies and percentages of samples at different scoring ranges. As shown in the figure, 14 cases out of 164 negatively expressed Ki-67, whilst the remaining showed positive expression of Ki-67 at variable percentages.

### Relationship between SP and Ki-67 expression levels and clinicopathological parameters

As shown in Table 2, significant associations were found between SP expression level and breast cancer molecular subtype (p = 0.002), TNM stage (p = 0.034), pN stage (p = 0.013), axillary lymph node metastasis (p = 0.004), ER and PR statuses (p<0.001) and history of DCIS (p = 0.009).

**Table 1. Demographic and clinicopathological characteristics of 164 female patients with breast cancer.**

| Variable | Total (n%) |
|---|:---:|
| **Age (Years)** | |
| Mean ± SD | 51.35 ± 11.2 |
| Range | 28–82 |
| **Breast cancer molecular subtype** | |
| Her2/neu-enriched | 42 (25.6) |
| Luminal A | 40 (24.4) |
| Luminal B (Triple positive) | 42 (25.6) |
| Triple negative (TNBC) | 40 (24.4) |
| **Grade** | |
| I | 8 (4.9) |
| II | 46 (28) |
| III | 110 (67.1) |
| **Tumor volume (cm$^3$)** | |
| Mean ± SD | 38.97 ± 67.41 |
| Range | 0.18–571.77 |
| **TNM Stage** | |
| I | 7 (4.3) |
| II | 45 (27.4) |
| III | 56 (34.1) |
| IV | 53 (32.3) |
| Undetermined | 3 (1.8) |
| **pT stage** | |
| pT1 | 13 (7.9) |
| pT2 | 88 (53.7) |
| pT3 | 47 (28.7) |
| pT4 | 16 (9.8) |
| **pN stage** | |
| pN0 | 44 (26.8) |
| pN1 | 39 (23.8) |
| pN2 | 33 (20.1) |
| pN3 | 43 (26.2) |
| Undetermined | 5 (3.0) |
| **Distant metastasis** | |
| M0 | 107 (65.2) |
| M1 | 54 (32.9) |
| Undetermined | 3 (1.8) |
| **Axillary lymph node metastasis** | |
| Negative | 45 (27.4) |
| Positive | 117 (71.3) |
| Undetermined | 2 (1.2) |
| **Lymphovascular invasion** | |
| Negative | 31 (18.9) |
| Positive | 94 (57.3) |
| Undetermined | 39 (23.8) |
| **ER status** | |
| Negative | 82 (50) |
| Positive | 82 (50) |

(*Continued*)

**Table 1.** (Continued)

| Variable | Total (n%) |
| --- | --- |
| **PR status** | |
| Negative | 82 (48.8) |
| Positive | 82 (51.2) |
| **Her2/neu status** | |
| Negative | 80 (48.8) |
| Positive | 84 (51.2) |
| **DCIS history** | |
| Absent | 27 (16.5) |
| Present | 113 (68.9) |
| Undetermined | 24 (14.6) |
| **Family history** | |
| No | 106 (64.6) |
| Yes | 29 (17.7) |
| Undetermined | 29 (17.7) |
| **Ki-67 (%)** | |
| Mean ± SD | 27.05 ± 26.86 |
| Range | 0–95 |
| **SP** | |
| Low | 36 (22) |
| High | 128 (78) |

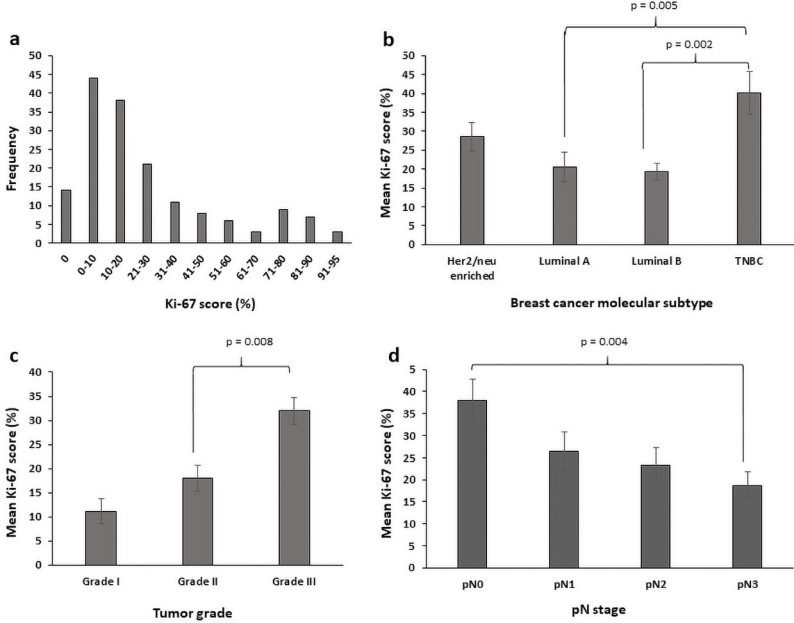

**Fig 2. Distribution of Ki-67 expression in breast cancer cases.** (a) Number of cases in each group with different Ki-67 scores. (b) Mean Ki-67 scores in different breast cancer molecular subtypes. (c) Mean Ki-67 scores in breast cancer cases of different grades. (d) Mean Ki-67 scores in breast cancer cases of different pN stages. TNBC, triple negative breast cancer. Tukey's post hoc multiple comparisons test was performed to identify significant differences between breast cancer subgroups.

**Table 2. Clinicopathological features of breast cancer patients in relation to Substance P (SP) expression.**

| Parameters | Low or no expression Percentage (%)/ Mean ± SD | High expression Percentage (%)/Mean ± SD | t-test coefficient/ Chi-square | p |
|---|---|---|---|---|
| **Age (Years)** | 52.33 ± 12.67 | 51.08 ± 10.80 | 0.592 | 0.554 |
| **Breast cancer molecular subtype** | | | 14.751 | **0.002** |
| Her2/neu-enriched | 13 (31.0) | 29 (69.0) | | |
| Luminal A | 4 (10.0) | 36 (90.0) | | |
| Luminal B (Triple positive) | 4 (9.5) | 38 (90.5) | | |
| Triple negative (TNBC) | 15 (37.5) | 25 (62.5) | | |
| **Tumor grade** | | | 0.752 | 0.687 |
| I | 1 (12.5) | 7 (87.5) | | |
| II | 9 (19.6) | 37 (80.4) | | |
| III | 26 (23.6) | 84 (76.4) | | |
| **Tumor volume (cm3)** | 48.47 ± 68.92 | 36.29 ± 67.02 | 0.957 | 0.340 |
| **TNM Stage** | | | 8.690 | **0.034** |
| I | 1 (14.3) | 6 (85.7) | | |
| II | 17 (37.8) | 28 (62.2) | | |
| III | 10 (17.9) | 46 (82.1) | | |
| IV | 8 (15.1) | 45 (84.9) | | |
| **pT stage** | | | 7.227 | 0.065 |
| pT1 | 3 (23.1) | 10 (76.9) | | |
| pT2 | 24 (27.3) | 64 (72.7) | | |
| pT3 | 4 (8.5) | 43 (91.5) | | |
| pT4 | 5 (31.3) | 11 (68.7) | | |
| **pN stage** | | | 10.795 | **0.013** |
| pN0 | 17 (38.6) | 27 (61.4) | | |
| pN1 | 6 (15.4) | 33 (84.6) | | |
| pN2 | 7 (21.2) | 26 (78.8) | | |
| pN3 | 5 (11.6) | 38 (88.4) | | |
| **Distant metastasis** | | | 2.665 | 0.074 |
| M0 | 28 (26.2) | 79 (73.8) | | |
| M1 | 8 (14.8) | 46 (85.2) | | |
| **Axillary lymph nodes metastasis** | | | 8.723 | **0.004** |
| Negative | 17 (37.8) | 28 (62.2) | | |
| Positive | 19 (16.2) | 98 (83.8) | | |
| **Lymphovascular invasion** | | | 3.285 | 0.063 |
| Negative | 10 (32.3) | 21 (67.7) | | |
| Positive | 16 (17.0) | 78 (83.0) | | |
| **ER status** | | | 14.236 | **<0.001** |
| Negative | 28 (34.1) | 54 (65.9) | | |
| Positive | 8 (9.8) | 74 (90.2) | | |
| **PR status** | | | 14.236 | **<0.001** |
| Negative | 28 (34.1) | 54 (65.9) | | |
| Positive | 8 (9.8) | 74 (90.2) | | |
| **Her2/neu status** | | | 0.295 | 0.361 |
| Negative | 19 (23.8) | 61 (76.2) | | |
| Positive | 17 (20.2) | 67 (79.8) | | |

(*Continued*)

**Table 2.** (Continued）

| Parameters | Low or no expression Percentage (%)/ Mean ± SD | High expression Percentage (%)/Mean ± SD | t-test coefficient/ Chi-square | p |
|---|---|---|---|---|
| **DCIS history** | | | 7.410 | **0.009** |
| Absent | 11 (40.7) | 16 (59.3) | | |
| Present | 19 (16.8) | 94 (83.2) | | |
| **Family History** | | | 0.000 | 0.609 |
| No | 22 (20.8) | 84 (79.2) | | |
| Yes | 6 (20.7) | 23 (79.3) | | |
| **Ki-67 (%)** | 26.69 ± 30.62 | 27.15 ± 25.83 | -0.089 | 0.929 |

In particular, low SP expression levels were seen in breast cancer tissues of TNBC subtype, TNM stage II and pN0, while high SP expression levels were seen in tissues of pT3 stage (Fig 3).

Ki-67 was analyzed as a continuous variable. The expression levels of Ki-67 had no significant association with age, tumor volume, TNM stage, distant metastasis, presence of lymphovascular invasion, Her2/neu status, history of DCIS, family history of breast cancer or SP expression levels. However, statistically significant differences were observed between the mean Ki-67 scores and molecular subtype (p = 0.001), grade (p = 0.003), pN stage (p = 0.007), axillary lymph node metastasis (p = 0.001), and ER and PR statuses (p <0.001); Table 3.

As shown in Fig 2b–2d, a Tukey post hoc multiple comparisons test revealed that the mean Ki-67 score was higher in the TNBC subtype (40.1 ± 35.4) compared to the luminal A (20.6 ± 24.9; p = 0.005) and the luminal B subtypes (19.3 ± 14.6; p = 0.002). There were no statistically significant differences between the Her2/neu-enriched subtype and any of the molecular subtypes, or between the luminal A and the luminal B subtypes (p >0.05). Additionally, the mean Ki-67 score in grade III tumors (31.9 ± 29.3) was significantly higher than that in grade II tumors (18.1 ± 18.3; p = 0.008), a near significant difference was found between grade

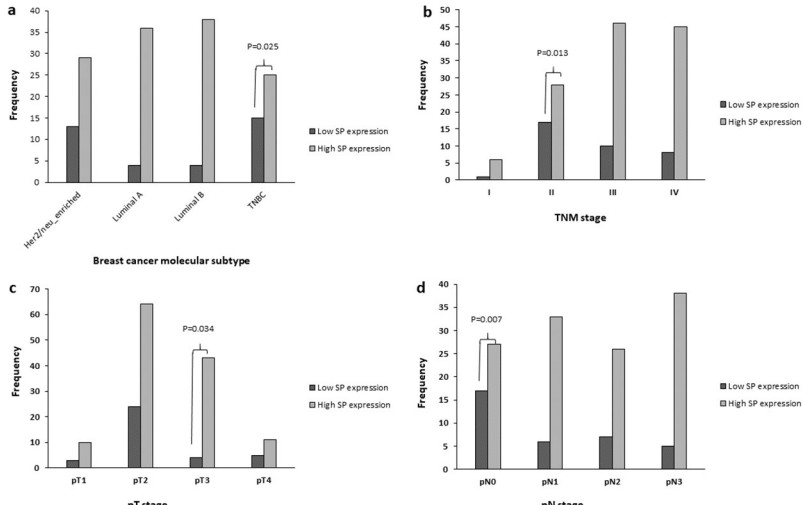

**Fig 3. Distribution of substance P (SP) expression in breast cancer cases.** (a) SP expression level in different breast cancer molecular subtypes. (b) SP expression level in breast cancer cases of different TNM stages. (c) SP expression level in breast cancer cases of different pT stages. (d) SP expression level in breast cancer cases of different pN stages. TNBC, triple negative breast cancer. Bonferroni Chi-Square residual analysis was performed to identify significant differences between breast cancer subgroups. Adjusted p-values were calculated as 0.05/n$_{tests}$.

**Table 3. Clinicopathological features of breast cancer patients in relation to Ki-67 expression.**

| Variable | Ki-67 mean score (% ± SD) | P |
|---|---|---|
| **Age (Years)** | 51.35 ± 11.2 | 0.607 |
| **Breast cancer molecular subtype** | | **0.001** |
| Her2/neu-enriched | 28.5 ± 24.5 | |
| Luminal A | 20.6 ± 24.9 | |
| Luminal B (Triple positive) | 19.3 ± 14.6 | |
| Triple negative (TNBC) | 40.1 ± 35.4 | |
| **Grade** | | **0.003** |
| I | 11.1 ± 7.4 | |
| II | 18.1 ± 18.3 | |
| III | 31.9 ± 29.3 | |
| **Tumor volume (cm³)** | 27.05 ± 26.856 | 0.721 |
| **TNM Stage** | | 0.114 |
| I | 44.9 ± 25.7 | |
| II | 31.2 ± 31.2 | |
| III | 22.4 ± 20.1 | |
| IV | 25.5 ± 28.8 | |
| **pT stage** | | 0.595 |
| pT1 | 30.7 ± 28.4 | |
| pT2 | 28.7 ± 29.6 | |
| pT3 | 25.5 ± 24.0 | |
| pT4 | 19.8 ± 16.0 | |
| **pN stage** | | **0.007** |
| pN0 | 37.9 ± 31.9 | |
| pN1 | 26.4 ± 27.6 | |
| pN2 | 23.2 ± 23.2 | |
| pN3 | 18.7 ± 20.0 | |
| **Distant metastasis** | | 0.665 |
| M0 | 27.5 ± 26.3 | |
| M1 | 25.5 ± 28.5 | |
| **Axillary lymph node metastasis** | | **0.001** |
| No | 37.9 ± 31.6 | |
| Yes | 22.7 ± 23.7 | |
| **Lymphovascular invasion** | | 0.938 |
| Negative | 26.8 ± 28.1 | |
| Positive | 27.2 ± 26.8 | |
| **ER status** | | **< 0.001** |
| Negative | 34.2 ± 30.7 | |
| Positive | 19.9 ± 20.2 | |
| **PR status** | 34.2 ± 30.7 | **< 0.001** |
| Negative | 19.9 ± 20.2 | |
| Positive | | |
| **Her-2/neu status** | 30.4 ± 31.9 | 0.125 |
| Negative | 23.9 ± 20.6 | |
| Positive | | |
| **DCIS history** | | 0.250 |
| Absent | 32.9 ± 30.7 | |
| Present | 26.2 ± 25.9 | |

(*Continued*)

**Table 3.** (Continued)

| Variable | Ki-67 mean score (% ± SD) | P |
|---|---|---|
| **Family history** | | 0.193 |
| No | 25.4 ± 26.9 | |
| Yes | 32.8 ± 28.6 | |
| **SP** | | 0.929 |
| Low | 26.69 ± 30.62 | |
| High | 27.15 ± 25.83 | |

I (11.1 ± 7.4) and grade III (p = 0.077), and no statistically significant difference was found between grade I and grade II (p = 0.765). Moreover, tumors with pN3 (18.7 ± 20.0) had a significantly lower mean Ki-67 score than tumors with pN0 (37.9 ± 31.9; p = 0.004). A near significant difference in mean Ki-67 score was found between tumors with pN0 and those with pN2 (23.2 ± 23.2; p = 0.074).

## Discussion

According to the 2016 updated staging of breast cancer by the American Joint Committee on Cancer (AJCC), TNM stage, tumor grade and expression of ER/PR and Her2/neu are the three main parameters utilized for prognostic staging of breast cancer, which is then used to shape the treatment plan [29]. Nevertheless, the considerable variability in the biological behavior of the different breast cancer subtypes necessitates better understanding of the molecular background of breast cancer through the process of tumorigenesis.

SP expression may represent a useful prognostic marker in breast cancer and a novel therapeutic target thereafter. For the first time, the expression of SP in the four molecular subtypes of human breast cancer, luminal A, Her/neu-enriched, luminal B (triple positive) and TNBC, to be described and the possible association with the proliferation index Ki-67 to be investigated. SP expression was observed in most of the investigated cases of breast cancer tissues and high expression levels were revealed in more than three quarters of them. High SP expression levels were shown previously in about 30% of the investigated breast cancer samples (34 cases) [19], and in about 70% of colorectal cancer cases [18].

SP expression was found to significantly associate with breast cancer molecular subtype. The frequency of high SP expression was as following: 29.7% were luminal B (triple positive) molecular subtype, 28.1% were luminal A, 22.7% were Her2/neu-enriched and 19.5% were triple negative. Additionally, there was a significant association between SP expression level and TNM stage (p = 0.034). The highest frequencies of high SP expression were among cases with advanced TNM stages III (36.8%) and IV (36%) as compared to stages I (4.8%) and II (22.4%). Thus, indicating that SP could represent a potent prognostic biomarker in breast cancer. Such association could not be proved in colorectal cancer [18], and was not previously investigated in breast cancer.

A statistically significant association was also found between SP expression and pN stage (p = 0.013) and axillary lymph node metastasis (p = 0.004), besides a near significant association with the presence of distant metastasis (p = 0.074) and a positive lymphovascular invasion of the tumor tissue (p = 0.063). These results support the previous data that reported a role of SP in promoting cancer metastasis through enhancing angiogenesis and the proliferation and migration of cancer cells [12, 16, 18, 20, 27, 28, 30–32].

Uncontrolled increased cellular proliferation is a hallmark of cancer which resulted in the development of therapies that are targeted to different proliferation markers such as endocrine

therapies represented by antiestrogen agents like tamoxifen [33]. Nevertheless, resistance to these agents is still an issue that may lead to cancer progression and death. Consequently, cell cycle targeted agents represented by selective cyclin dependent kinase (CDK4/6) inhibitors were approved as adjunct therapy to endocrine agents to overcome the resistance and improve the prognosis of patients [34]. However, eventual resistance may also occur while the mechanisms are not yet fully understood [35, 36]. Better understanding of the molecular mechanisms that confer resistance to the current therapies will aid in the identification of new markers that may represent novel therapeutic targets in breast cancer. The effect of SP on cellular proliferation was previously observed in breast cancer cell lines where the use of anti-SP antibodies resulted in the downregulation of EGFR and Her2/neu [27]. Conversely, treatment of breast cancer cell lines with SP resulted in overexpression of Her2/neu and EGFR and affected the response to anti-EGFR and anti-Her2 agents [15]. Our results showed a significant positive association between SP expression and ER and PR statuses ($p<0.001$) but not with Her2/neu status or Ki-67 expression level. Our results indicate a role of SP in breast cancer proliferation and progression through interacting with hormone receptors. A study by Villablanca et al., reported a stimulatory effect of 17β estradiol on NK1R gene expression levels which was also associated with an increased specific binding of SP to its receptor, thus claiming a hormonal control of NK1R expression and SP function [37].

Early diagnosis is the best strategy to combat cancer. Ductal carcinoma in situ (DCIS), a noninvasive nonobligate precursor of breast cancer, accounts for about 25% of all breast cancer cases [38]. Nevertheless, there is still controversy about the actual percentage that can transform into invasive breast cancer and about the time to progression [38]. Therefore, biomarkers to determine the potentiality of DCIS to progress into invasive cancer are required. These progression biomarkers will also allow for proper monitoring of DCIS patients and personalization of therapy. In our study we found a significant association between SP expression and personal history of DCIS, where 85.5% of patients who had a DCIS history expressed SP at high levels (p = 0.009).

The clinical significance of Ki-67 in our cohort was also evaluated. The mean score of Ki-67 in our cohort was about 27% which is close to that reported by previous studies [39–42].

In the analyses, Ki-67 was evaluated as a continuous variable and we found a significant association with breast cancer molecular subtype, tumor grade, pN stage, axillary node metastasis and ER and PR statuses. Statistically significant higher mean value of Ki-67 expression was found in the TNBC subtype as compared to luminal A (p = 0.005) and luminal B (triple positive) subtypes (p = 0.002). Differential expression of Ki-67 among breast cancer molecular subtypes was previously investigated in different studies, where high expression level among cases of TNBC subtype was reported [42–46]. Moreover, our result that show a significantly higher Ki-67 expression levels among cases of high grade (grade III) is in accordance with the results of previous studies [42, 43, 45–47].

Low pN stage and negative axillary lymph node metastasis were associated with lower Ki-67 expression levels. Controversial outcomes exist in literature regarding the association between lymph node metastasis and Ki-67 expression level. Findings that are opposite to ours were reported by some studies [42, 48, 49], and no association could be found in other studies [45, 46, 50].

ER and PR statuses are universal predictive and prognostic biomarkers in breast cancer [51], however disparate results are available in literature regarding their relationship with Ki-67 expression level [42, 48, 52–55]. In our study, cases with ER negative or PR negative status had higher mean Ki-67 scores. Our results are in accordance with those revealed by some others that investigated Ki-67 as a continuous variable in breast cancer [40, 42, 46, 54].

We are aware that our study has some limitations which include the small sample size and the unavailability of data on survival and recurrence status. Therefore, we recommend prospective large-scale future studies to evaluate the prognostic and predictive significance of SP in combination with Ki-67 in breast cancer.

As a conclusion, our results show that higher expression level of SP are associated with TNBC breast cancer molecular subtype, TNM stage II, pT3, pN0, positive axillary lymph node metastasis, positive ER and PR statuses and positive personal history of DCIS. Additionally, higher Ki-67 expression levels were seen in cancer tissues of TNBC molecular subtype, tumor grade III, pN0 stage, positive axillary lymph node metastasis and positive ER and PR statuses. However, we could not find a relationship between SP and Ki-67 expression levels. Future prospective studies with larger sample size are recommended to study the prognostic value of immunohistochemical expression of SP in breast cancer and the ability to utilize it for subclassification of patients for better stratification to therapy.

## Supporting information

**S1 File. The complete dataset of clinical, pathological, and staining data of breast cancer patients.**
(XLSX)

## Author Contributions

**Conceptualization:** Maha S. Al-Keilani, Mohammad A. Alqudah, Asem M. Alkhateeb.

**Data curation:** Maha S. Al-Keilani, Rana I. Elstaty.

**Formal analysis:** Maha S. Al-Keilani, Rana I. Elstaty, Mohammad A. Alqudah.

**Funding acquisition:** Maha S. Al-Keilani.

**Investigation:** Maha S. Al-Keilani, Mohammad A. Alqudah.

**Methodology:** Maha S. Al-Keilani, Mohammad A. Alqudah.

**Project administration:** Maha S. Al-Keilani.

**Resources:** Mohammad A. Alqudah.

**Supervision:** Maha S. Al-Keilani, Asem M. Alkhateeb.

**Validation:** Maha S. Al-Keilani, Mohammad A. Alqudah, Asem M. Alkhateeb.

**Visualization:** Maha S. Al-Keilani, Asem M. Alkhateeb.

**Writing – original draft:** Maha S. Al-Keilani.

**Writing – review & editing:** Maha S. Al-Keilani, Rana I. Elstaty, Mohammad A. Alqudah, Asem M. Alkhateeb.

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
