## [Decision Letter · Decision Letter 0]

28 Apr 2021

PONE-D-21-03600

Immunohistochemical Expression of Substance P in Breast Cancer and its Association with Prognostic Parameters and Ki-67 Index.

PLOS ONE

Dear Dr. El Keillani,

Thank you for submitting your manuscript to PLOS ONE. After careful consideration, we feel that it has merit but does not fully meet PLOS ONE’s publication criteria as it currently stands. Therefore, we invite you to submit a revised version of the manuscript that addresses the points raised during the review process.

We look forward to receiving your revised manuscript.

Kind regards,

Vahit Özmen, MD, FACS

Professor of Surgery

Academic Editor

PLOS ONE

Journal Requirements:

4a) If there are ethical or legal restrictions on sharing a de-identified data set, please explain them in detail (e.g., data contain potentially identifying or sensitive patient information) and who has imposed them (e.g., an ethics committee). Please also provide contact information for a data access committee, ethics committee, or other institutional body to which data requests may be sent.

4b) If there are no restrictions, please upload the minimal anonymized data set necessary to replicate your study findings as either Supporting Information files or to a stable, public repository and provide us with the relevant URLs, DOIs, or accession numbers. Please see http://www.bmj.com/content/340/bmj.c181.long for guidelines on how to de-identify and prepare clinical data for publication. For a list of acceptable repositories, please see http://journals.plos.org/plosone/s/data-availability#loc-recommended-repositories.

Reviewers' comments:

Reviewer's Responses to Questions

**Comments to the Author**

1. Is the manuscript technically sound, and do the data support the conclusions?

Reviewer #1: Partly

Reviewer #2: Yes

2. Has the statistical analysis been performed appropriately and rigorously? 

Reviewer #1: I Don't Know

Reviewer #2: Yes

3. Have the authors made all data underlying the findings in their manuscript fully available?

Reviewer #1: Yes

Reviewer #2: Yes

4. Is the manuscript presented in an intelligible fashion and written in standard English?

Reviewer #1: Yes

Reviewer #2: Yes

5. Review Comments to the Author

Reviewer #1: - Please include the approval number of your instutional review board and the reason why you did not obtain form of consent from patients.

- In line 49 "targeted" would be better instead of "biological".

- In line 80 what do you mean by "depth of infiltration"?

- How did you construct your tissue microarrays? What was the size of each core, how many cores did you take from each case? Which instrument did you use?

- For ER and PR positivity what is your trashold? How did you evaluate HER2 by immunohistochemistry or FISH or both of them? Though you mentioned that you obtained thesse results pathology reports, please give some more detailed information.

- In line 161 you wrote that you have whown SP expression levels in Table 1 but they were shown them in table 2

- In line 165 what do you mean by "negative expression of Ki67"?

- In line 203 it would be better if you write "prognostic staging" instead of "full staging" with mentioning the date of last AJCC.

- You did not give detailed histroy of DCIS of your cases. Then you suggest that there is high expression of SP in women with history of DCIS. We do not know the time period between diagnoisis of DCIS and invasive carcinoma, the site of DCIS, the type of treatment? It is quite speculative to draw such a conclusion.

- Please re-write last sentences of discussion of your manuscript based on your results. As you have not evaluated the relation between SP expression and prognosis and there is nothing about the diagnosis of breast cancer as well. The last sentences is not relevant to your findings.

Reviewer #2: Substance P in breast cancer is a relatively under-researched area. The article is interesting with this aspect. However, some correction is required for it to be better.

Line 80. "Clinicopathological data including age, gender, tumor volume, degree of histological 80 differentiation (well / moderate / poor, WHO), depth of infiltration, staging, and status of lymph 81 nodes and distant metastasis, and hormone receptor (ER and PR) and Her2 / neu statuses were 82 retrospectively collected from patients' medical charts. " I could not understand what "depth of infiltration" is used for.

Lİne 161. Table 1 should be changed to table 2.

Figure 2 should be omitted.

Table 2 desing is done incorrectly. The percentage of substance P level for each molecular subtype should be given in the table. For example Her2 / neu enriched; 13/42 (30.95%), 29/42 (69.05%). The table should be rearranged as shown in the example. This error continues with stage, grade and other parameters. Additionally, the correction regarding these numbers (percentage etc.) should also be made in the discussion.

There is a difference in substance P expression level between which molecular subtypes are shown in Table 2. The p value for each molecular type should be given separately.

Is there any relationship between Ki67 and substance P? I could not see it in the text and in the summary. However, in the study with the title it is written that investigated this relationship. "Immunohistochemical Expression of Substance P in Breast Cancer and its Association with Prognostic Parameters and Ki-67 Index."

In the conclusion section, it should be written clearly in which molecular subtypes and at which stage the SP and ki67 levels are higher. The same is necessary for its relationship with ER, PR and lymph node metastasis. With this point of view, the summary should also be rearranged.

6. PLOS authors have the option to publish the peer review history of their article (what does this mean?). If published, this will include your full peer review and any attached files.

Reviewer #1: No

Reviewer #2: No

---

## [Author Response · Author response to Decision Letter 0]

16 May 2021

May 17th, 2021

PONE-D-21-03600

Dear Editor,

We want to thank you and the reviewers for the generous comments on the manuscript titled “Immunohistochemical Expression of Substance P in Breast Cancer and its Association with Prognostic Parameters and Ki-67 Index”. We have edited the manuscript to address all the comments.

Below please find our responses:

A. Journal Requirements:

Response: Done

Response: Done

Response: The phrase that refers to the “data not shown” has been removed as it is not essential for this article.

4a) If there are ethical or legal restrictions on sharing a de-identified data set, please explain them in detail (e.g., data contain potentially identifying or sensitive patient information) and who has imposed them (e.g., an ethics committee). Please also provide contact information for a data access committee, ethics committee, or other institutional body to which data requests may be sent.

4b) If there are no restrictions, please upload the minimal anonymized data set necessary to replicate your study findings as either Supporting Information files or to a stable, public repository and provide us with the relevant URLs, DOIs, or accession numbers. Please see http://www.bmj.com/content/340/bmj.c181.long for guidelines on how to de-identify and prepare clinical data for publication. For a list of acceptable repositories, please see http://journals.plos.org/plosone/s/data-availability#loc-recommended-repositories.

Response: The data has been uploaded as supplementary file (S1 File). This is notified in the statistical analysis subsection of materials and methods line 147 of the revised manuscript.

B. Reviewers' comments:

Reviewer #1:

1- Please include the approval number of your institutional review board and the reason why you did not obtain form of consent from patients.

Response: The IRB approval reference number has been added as requested under a new subsection in the materials and methods section (lines 86-90). Due to the retrospective nature of the study and that all the data were fully anonymized, a consent form was waived by the IRB.

2- In line 49 "targeted" would be better instead of "biological".

Response: “biological” has been replaced with “targeted” as recommended.

3- In line 80 what do you mean by "depth of infiltration"?

Response: How deep tumor cells invade into the tissue, which is considered one of the prognostic factors in breast cancer.

4- How did you construct your tissue microarrays? What was the size of each core, how many cores did you take from each case? Which instrument did you use?

Response: More details were added to the TMA subsection (Lines 94-105)

5- For ER and PR positivity what is your threshold? How did you evaluate HER2 by immunohistochemistry or FISH or both of them? Though you mentioned that you obtained these results pathology reports, please give some more detailed information.

Response: The cut-off value for ER and PR positivity was set at 5%.

Her2/neu status was evaluated according to the guidelines of IHC. FISH for Her2/neu status was not performed at the pathology department at KAUH. The guidelines consider 0 or +1 as negative, +3 as positive. All cases of +2 were sent to a more specialized center to do FISH and the results received were added to the pathology report as either positive or negative.

The above information has been added to the materials and methods section lines 80-84.

6- In line 161 you wrote that you have shown SP expression levels in Table 1 but they were shown them in table 2

Response: The information are in Table 1 as cited, but for unknown reason the remaining of the table was omitted from the manuscript file. Thus we added the remaining of the table to the revised file.

7- In line 165 what do you mean by "negative expression of Ki67"?

Response: Negative expression means no staining for Ki-67 (score 0)

8- In line 203 it would be better if you write "prognostic staging" instead of "full staging" with mentioning the date of last AJCC.

Response: The term “full staging” was replaced by “prognostic staging” as recommended.

The date of the last AJCC was added as recommended.

9- You did not give detailed history of DCIS of your cases. Then you suggest that there is high expression of SP in women with history of DCIS. We do not know the time period between diagnosis of DCIS and invasive carcinoma, the site of DCIS, the type of treatment? It is quite speculative to draw such a conclusion.

Response: In almost all cases, DCIS is usually diagnosed at the same time in the same biopsy with invasive cancer. But in minority of cases, it is diagnosed alone. In such cases an immediate action to proceed with surgery is taken and duration does not exceed 2-3 weeks between both procedures 

10- Please re-write last sentences of discussion of your manuscript based on your results. As you have not evaluated the relation between SP expression and prognosis and there is nothing about the diagnosis of breast cancer as well. The last sentences is not relevant to your findings.

Response: The sentences have been rewritten as recommended. (lines 309-312)

Reviewer #2: 

Substance P in breast cancer is a relatively under-researched area. The article is interesting with this aspect. However, some correction is required for it to be better.

1- Line 80. "Clinicopathological data including age, gender, tumor volume, degree of histological 80 differentiation (well / moderate / poor, WHO), depth of infiltration, staging, and status of lymph nodes and distant metastasis, and hormone receptor (ER and PR) and Her2 / neu statuses were retrospectively collected from patients' medical charts. " I could not understand what "depth of infiltration" is used for.

Response: How deep tumor cells invade into the tissue, which is considered one of the prognostic factors in breast cancer.

2- Line 161. Table 1 should be changed to table 2.

Response: The information is in Table 1 as cited, but for unknown reason the remaining of the table was omitted from the manuscript file. Thus we added the remaining of the table to the revised file.

3- Figure 2 should be omitted.

Response: The figure has been omitted as recommended.

4- Table 2 design is done incorrectly. The percentage of substance P level for each molecular subtype should be given in the table. For example Her2 / neu enriched; 13/42 (30.95%), 29/42 (69.05%). The table should be rearranged as shown in the example. This error continues with stage, grade and other parameters. Additionally, the correction regarding these numbers (percentage etc.) should also be made in the discussion.

Response: Table 2 has been adjusted as recommended. The paragraph describing the numbers has been omitted as all the data are available in the table.

5- There is a difference in substance P expression level between which molecular subtypes are shown in Table 2. The p value for each molecular type should be given separately.

Response: Post hoc analysis by applying Bonferroni correction was performed. A figure to represent the results of the analysis are shown in an added figure (Fig. 3)

6- Is there any relationship between Ki67 and substance P? I could not see it in the text and in the summary. However, in the study with the title it is written that investigated this relationship. "Immunohistochemical Expression of Substance P in Breast Cancer and its Association with Prognostic Parameters and Ki-67 Index."

Response: No significant relationship was found between SP and Ki-67 expression levels. This was mentioned in the results section (revised manuscript; line 203), and discussion section (revised manuscript; lines 264-265), and we added a sentence to the conclusion section (line 309).

7- In the conclusion section, it should be written clearly in which molecular subtypes and at which stage the SP and ki67 levels are higher. The same is necessary for its relationship with ER, PR and lymph node metastasis. With this point of view, the summary should also be rearranged.

Response: Done.

---

## [Editor Report · Decision Letter 1]

19 May 2021

Immunohistochemical expression of substance P in breast cancer and its association with prognostic parameters and Ki-67 index.

PONE-D-21-03600R1

Dear Dr. Al-Keilani,

We’re pleased to inform you that your manuscript has been judged scientifically suitable for publication and will be formally accepted for publication once it meets all outstanding technical requirements.

Kind regards,

Vahit Özmen, MD, FACS

Professor of Surgery

Academic Editor

PLOS ONE
---

## [Editor Report · Acceptance letter]

24 May 2021

PONE-D-21-03600R1 

Immunohistochemical expression of substance P in breast cancer and its association with prognostic parameters and Ki-67 index. 

Dear Dr. Al-Keilani:

I'm pleased to inform you that your manuscript has been deemed suitable for publication in PLOS ONE. Congratulations! Your manuscript is now with our production department. 

Kind regards, 

on behalf of

Dr. Vahit Özmen 

Academic Editor

PLOS ONE